# Running modulates primate and rodent visual cortex differently

John P Liska[1†], Declan P Rowley[1,2†], Trevor Thai Kim Nguyen[1],
Jens-Oliver Muthmann[1], Daniel A Butts[3], Jacob Yates[4*‡§], Alexander C Huk[1,2‡#]

[1]Departments of Neuroscience and Psychology, Center for Perceptual Systems, Institute for Neuroscience, The University of Texas at Austin, Austin, United States; [2]Departments of Ophthalmology and Psychiatry & Biobehavioral Sciences, Fuster Laboratory for Cognitive Neuroscience, UCLA, Los Angeles, United States; [3]Department of Biology and Program in Neuroscience and Cognitive Science, University of Maryland, College Park, United States; [4]Herbert Wertheim School of Optometry and Vision Science, University of California, Berkeley, Berkeley, United States

*For correspondence:
yates@berkeley.edu

[†]These authors contributed equally to this work
[‡]These authors also contributed equally to this work

Present address: [§]Herbert Wertheim School of Optometry and Vision Science, Berkeley, United States; [#]Fuster Laboratory for Cognitive Neuroscience, UCLA, Los Angeles, United States

Competing interest: The authors declare that no competing interests exist.

## eLife assessment

This **important** work advances our understanding of the differences in locomotion-induced modulation in primate and rodent visual cortexes and underlines the significant contribution cross-species comparisons make to investigating brain function. The evidence in support of these differences across species is **convincing**. This work will be of broad interest to neuroscientists.

## Abstract

When mice run, activity in their primary visual cortex (V1) is strongly modulated. This observation has altered conceptions of a brain region assumed to be a passive image processor. Extensive work has followed to dissect the circuits and functions of running-correlated modulation. However, it remains unclear whether visual processing in primates might similarly change during locomotion. We therefore measured V1 activity in marmosets while they viewed stimuli on a treadmill. In contrast to mouse, running-correlated modulations of marmoset V1 were small and tended to be slightly suppressive. Population-level analyses revealed trial-to-trial fluctuations of shared gain across V1 in both species, but while strongly correlated with running in mice, gain modulations were smaller and more often negatively correlated with running in marmosets. Thus, population-wide fluctuations of V1 may reflect a common feature of mammalian visual cortical function, but important quantitative differences point to distinct consequences for the relation between vision and action in primates versus rodents.

## Introduction

Sensation and action are traditionally thought to involve separate brain circuits serving distinct functions: activity in early sensory areas is driven nearly exclusively by the corresponding sensory input, whereas activity in motor areas is exclusively related to movement. Recent work in mice, a major mammalian model system in neuroscience, has called for a re-evaluation of this distinction, given recent demonstrations that activity in mouse primary visual cortex (V1) depends as much on whether the mouse is running or stationary as on what visual stimulus is shown (*Niell and Stryker, 2010*). Neurons in V1 of virtually all mammals are selective for simple image features, a presumably critical early step of image processing that continues throughout a hierarchy of visual brain areas (*Rosa and Krubitzer, 1999*; *Felleman and Van Essen, 1991*), and this is true of mice as well (*Niell and Stryker,*

*2008*). The observation that running modulates (mouse) V1 of a comparable magnitude to the visually driven activity has motivated substantial effort in the field to understand the biological mechanisms and functional consequences of this powerful interaction between sensation and action (*Keller et al., 2012*; *Vinck et al., 2015*; *Bennett et al., 2013*; *Saleem et al., 2013*; *Erisken et al., 2014*; *Reimer et al., 2014*; *Pakan et al., 2016*; *Polack et al., 2013*; *Fu et al., 2014*; *Mineault et al., 2016*; *Ayaz et al., 2013*; *Christensen and Pillow, 2022*; *Dipoppa et al., 2018*).

However, these observations have all been made in rodents; similar measurements have not been made in primates. Although rodents certainly rely on vision for important behaviors (*Hoy et al., 2016*; *Yilmaz and Meister, 2013*), primates are more fundamentally visual organisms, with exquisite acuity and specialized functional characteristics such as foveas and corresponding high-resolution representations of the central visual field in V1 (*Land and Fernald, 1992*), in addition to a larger network of areas involved in vision (*Felleman and Van Essen, 1991*). While experiments that allow subjects to run while viewing visual stimuli may now be commonplace in mice, analogous experiments in nonhuman primates have remained technically daunting. It has thus remained unclear whether the large effect of running on early visual processing is a general property of mammalian brains revealed by work in mice or whether the early stages of primate visual processing are less affected by nonvisual factors. Here, we fill this major gap in cross-species understanding by taking advantage of the relatively small size and peaceable nature of the common marmoset (*Callithrix jacchus*), which allowed us to have animals on a custom-designed treadmill and to use high-channel-count electrode arrays, including Neuropixels. Our comparative study fits into a larger emerging enterprise to assess whether substantial signals due to animal movements affect sensory processing similarly in rodents and primates (*Talluri et al., 2023*; *Stringer et al., 2019*; *Musall et al., 2019*).

## Results

We tested for running-based modulations in V1 of the common marmoset, a highly visual new world primate. Marmosets were head-fixed, placed on a wheel-based treadmill suited to their arboreal nature (*Figure 1a*), and alternated between running and not running while we presented various visual stimuli designed to assess the properties and responsiveness of V1 neurons (*Figure 1b*). We recorded from foveal and parafoveal neurons in two marmosets (using chronically implanted N-Form 3D electrode arrays), and in one marmoset were also able to simultaneously record from both foveal and peripheral V1 (using Neuropixels 1.0 probes). To support precise comparison to rodent V1, we used the same analysis pipeline on a publicly available mouse dataset that used matching stimuli in a treadmill paradigm (https://portal.brain-map.org/explore/circuits/visual-coding-neuropixels). This let us perform direct quantitative and statistical comparisons of the effects of running on V1 activity in a rodent and a primate.

First, we mapped the receptive fields of marmoset V1 neurons using reverse-correlation techniques adapted to free-viewing (*Yates et al., 2021*) while we measured gaze using a video-based eyetracker (*Figure 1c*). In V1 of both marmosets, we found receptive fields within the central few degrees of vision, with sizes expected at those eccentricities (1–5°, *Figure 1f*, blue and green; these can be compared to those in mouse, *Figure 1e*). As expected for primary visual cortex, marmoset V1 (both well-isolated single units and well-tuned multi-unit clusters) responded robustly to oriented gratings and exhibited orientation- (and sometimes direction-) selectivity (*Yu and Rosa, 2014*; *Sengpiel et al., 1996*), similar to that in the mouse V1 dataset (*Figure 1g and h*). Orientation tuning spanned a range from weak to strong tuning, with many units exhibiting strong and conventional tuning curves (*Figure 1i and j*).

As a first test for effects of running on V1 activity, we assessed whether running speed was correlated with aggregate V1 activity by comparing the time series of these variables throughout each session. In the mouse, such modulations are easily visually evident when inspecting the time series of neural activity and running: when the mouse runs, V1 spiking often increases substantially. *Figure 2a and b* shows example sessions with the maximal and median amounts of correlation between the time series of running speed and a generic low-dimensional representation of the population activity (the first principal component [PC] of the simultaneously recorded V1 trial spike counts, see 'Materials and methods'). This correlation could be seen when running/not running alternated on slow (*Figure 2a*) or fast (*Figure 2b*) time scales.

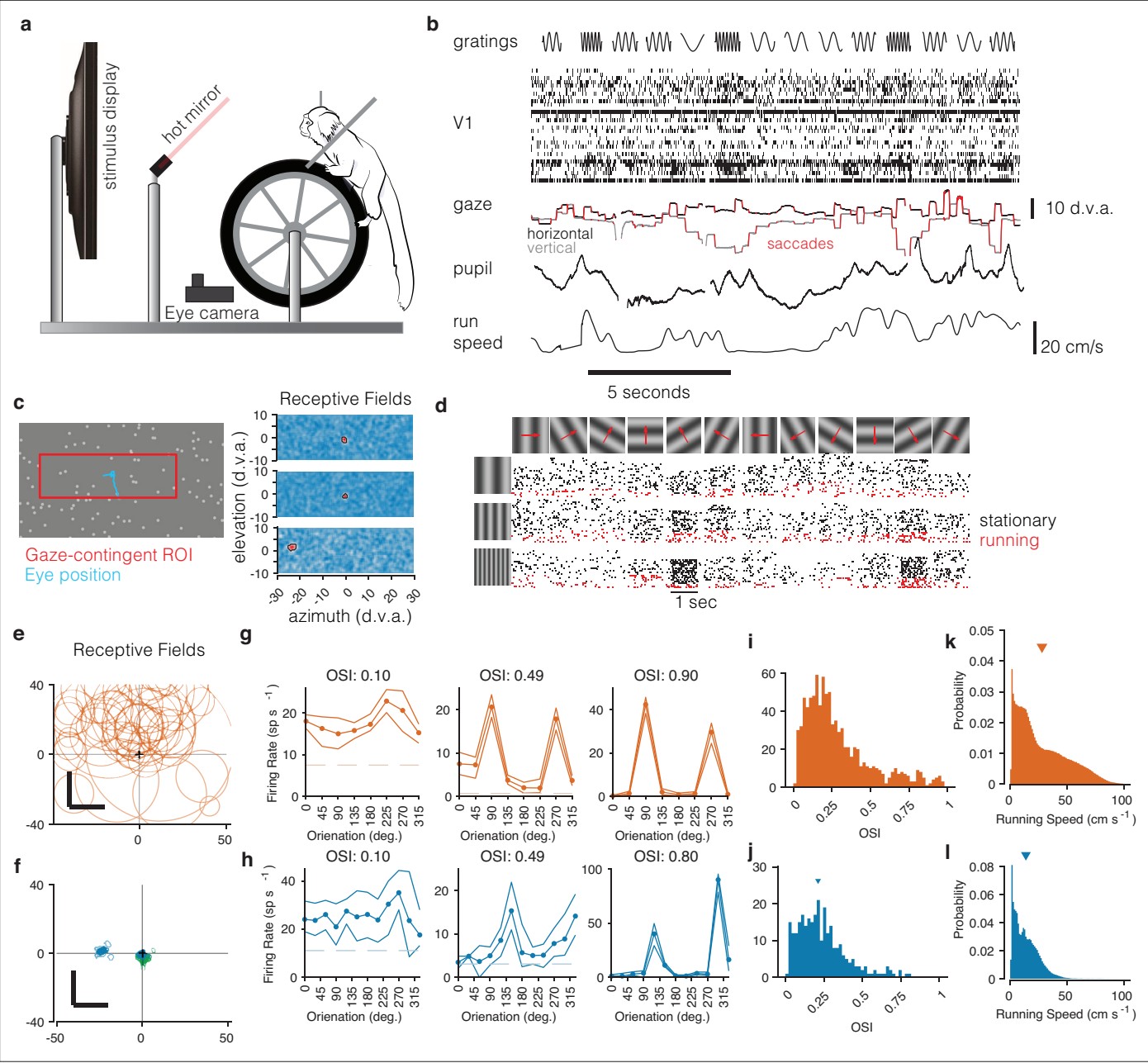

**Figure 1.** Recording from marmoset V1 during active locomotion. (**a**) Apparatus for recording from marmoset V1 while presenting visual stimuli on a high-resolution display, monitoring gaze using an eye tracker, on a toroidal treadmill that allowed the marmoset to run or not run. (**b**) Schematic example of variables of interest. Visual stimuli were presented (top row). Rasters show activity from a V1 array (second row). Gaze was monitored (third row, x and y time series plotted in black and gray), saccades were detected (red), and pupil size was also measured (fourth row). Running speed was measured using a rotary encoder attached to the treadmill (fifth row). (**c**) Before the main experiments, receptive fields (RFs) were mapped using sparse noise (*Yates et al., 2021*). The array of pseudocolor images shows three examples of V1 RFs (two foveal and one peripheral neuron). (**d**) Main experiment involved presenting full-field sinusoidal gratings that drifted in one of 12 directions (top row), at a variety of spatial frequencies (vertical axis at left). Rasters show example V1 activity during stimulus presentations when running (red) or stationary (black). (**e**) Summary of RF locations in the mouse dataset (orange, top), and (**f**) our data from marmosets (blue and green, bottom). In both marmosets, we recorded from a portion of V1 accessible at the dorsal surface of the brain using chronically implanted arrays, which yielded neurons with foveal RFs (green RFs). We also recorded from one marmoset using Neuropixels arrays, allowing us to simultaneously access both peripheral and foveal V1 (blue RFs; peripheral units are analyzed later/separately, see text). (**g**) Examples of mouse V1 orientation tuning curves, for cells with weak, moderate, and strong orientation tuning. (**h**) Same, for marmoset V1. (**i, j**) Histograms of orientation-selectivity indices (OSIs) for mice (**i**) and marmosets (**j**). Marmoset OSIs, likely lower than previously reported because we used full-field stimuli not optimized to the spatial frequency tuning of each neuron, and which likely recruited surround suppression. Regardless, the marmoset V1 neurons had strong visual responses and qualitatively conventional tuning. (**k, l**) Running speeds in mice (**k**) and marmosets (**l**). Marmosets were acclimated to the treadmill and motivated to run with fluid rewards yoked to traveling a criterion distance.

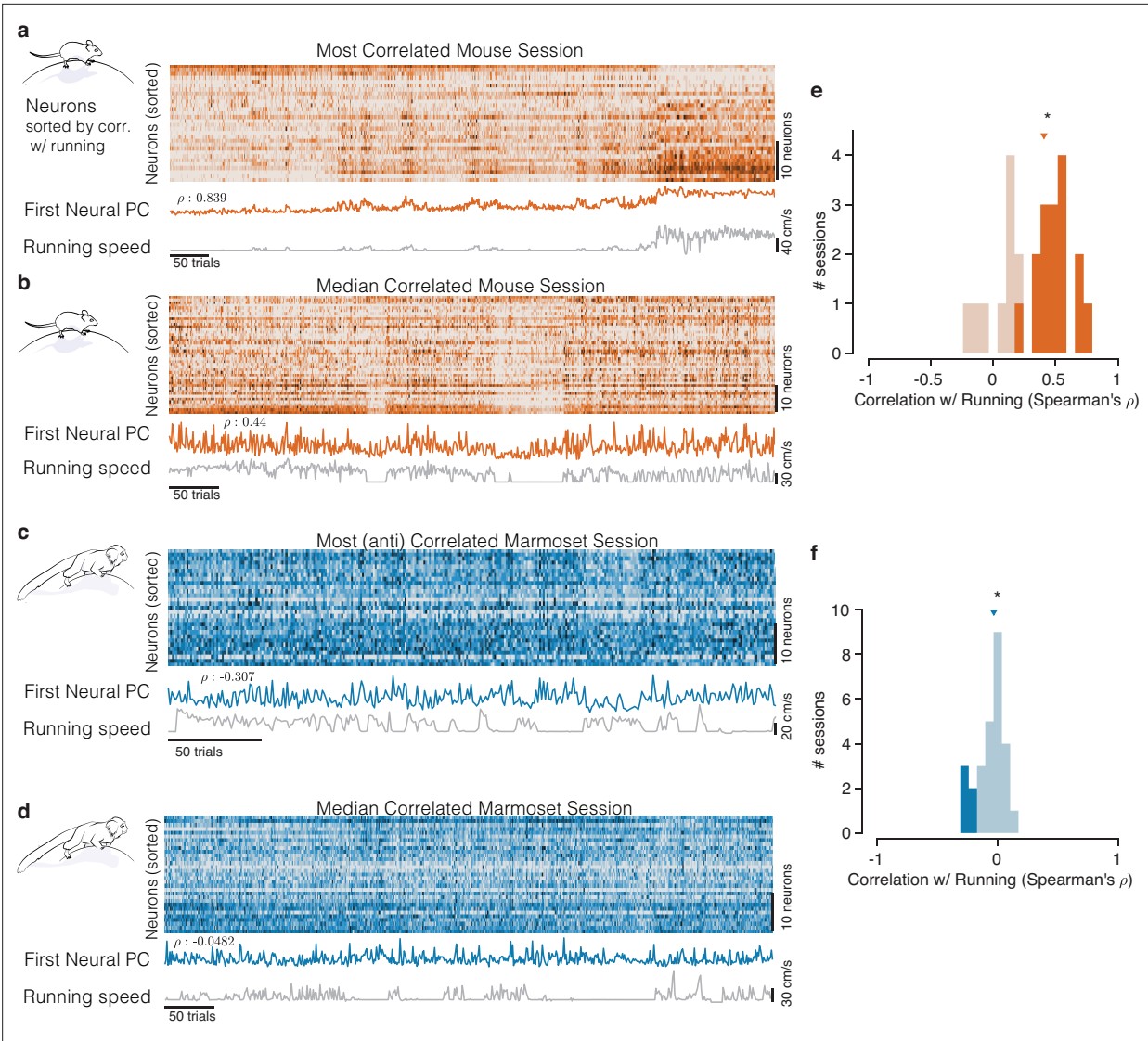

**Figure 2.** Mice and marmosets exhibit different correlations between V1 activity and running speed. (**a**) Mice show visually compelling correlations between V1 trial spike counts and running speed. Example session with the highest correlation between running and V1 activity. Raster at top shows spiking activity of all mouse V1 neurons recorded. Population activity is summarized below the raster as the first principal component of the V1 array activity ('First Neural PC', orange trace); running speed is plotted underneath it on the same time axis (gray trace). Clearly, the two curves are highly similar. (**b**) Same, for an example mouse session chosen to have the median correlation between running and V1 activity. In this example, the modulations of running speed and neural activity rise and fall together on a faster time scale than in the example in (**a**). (**c, d**) Marmosets show smaller, and typically negative, correlations between V1 spiking activity and running speed. Format same as the mouse data in (**a, b**), with example sessions chosen to show the maximal and the median correlations between V1 activity and running speed. The (anti-correlated) similarity between V1 activity (First Neural PC) and Running Speed curves is harder to discern in the marmoset. (**e, f**) Correlations between V1 activity and running in the mouse (**e**) had a median >0 (median = 0.407, p=$9.04 \times 10^{-5}$, stat = 308, n = 25, Mann–Whitney $U$ test), and many individual sessions had significant correlations with running (filled bars), and all such significant sessions had positive correlations (with significance determined via permutation to remove effects of autocorrelation; **Harris, 2021**). In the marmosets (**f**), the distributions of correlations were slightly but reliably negative (median = −0.033, p=0.034, stat = 101, n = 27, Mann–Whitney $U$ test), and all significantly modulated individual sessions exhibited negative correlations (5/27).

A starkly different impression comes from visual inspection of the relationship between running and the activity of marmoset V1 neurons representing the central visual field. Any relation between V1 activity and running appears considerably smaller. In examples showing the maximal and median relationships between running and V1 activity (**Figure 2c and d**), V1 activity did not track running speed as clearly, although the activity did tend to increase when the monkey stopped running, explaining the modest negative correlations.

We then quantified the relationship between the timecourses of aggregate V1 activity and running across all experiments on a session-by-session basis, in both species. For mice, this confirmed a strong positive correlation (*Figure 2e*; median = 0.407, n = 25, p=$9.04 \times 10^{-5}$, stat = 308, Mann–Whitney *U* test). For marmosets, the distribution of correlations between V1 activity and running was subtly but reliably negative (*Figure 2f*, median = −0.033, p=0.034, stat = 101, n = 27, Mann–Whitney *U* test). Most importantly, the correlation between V1 activity and running was significantly different between the two species (p=$6.93 \times 10^{-7}$, stat = 934, Mann–Whitney *U* test). This session-level analysis confirmed that running modulations in mice are large and mostly reflect increases in response. In contrast, running modulations in marmoset foveal V1 are small, and if anything, reflect slight reductions in activity.

To perform additional quantitative tests at the level of individual V1 units, we divvied up each unit's spiking responses to drifting gratings based on whether or not the animal was running (*Figure 3*). This analysis confirmed, in mouse, a tendency for large response increases during running to both the preferred orientation stimulus (*Figure 3a*, geometric mean ratio [running/stationary] = 1.523, 95% CI [1.469, 1.579], n = 743 tuned units) and to all visual stimuli (*Figure 3b*, 1.402 [1.365, 1.440], n = 1168). Many individual units had significant running modulations and were more often increases rather than decreases (803/1168 [69%] increased firing rate and 115/1168 [10%] decreased, bootstrapped *t*-test). In marmoset V1, there was again a modest decrease evident in the response to the preferred stimulus (*Figure 3c*; geometric mean ratio [running/stationary] = 0.899, 95% CI [0.851, 0.949], n = 228 tuned units). Not even modest suppression was evident in responses aggregated across all stimuli (*Figure 3d*, 1.011 [0.995, 1.027], n = 786). The number of significantly modulated units was relatively small and was more balanced between decreases and increases in firing rate (172/786 [22%] increased and 161/786 [20%] decreased, bootstrapped *t*-test). Because we performed quantitative comparisons on subsets of the data for which the stimuli were nearly identical across species, and used the same data analysis code to calculate response metrics, these analyses solidly confirm a substantial difference between the form of running modulations of V1 activity in mouse versus marmoset (log ratio of running:stationary was significantly different between mouse and marmoset for all units: p=$6.62 \times 10^{-99}$, stat = 1399874, Mann–Whitney *U* test, and tuned units: p=$4.69 \times 10^{-57}$, stat = 4030135). Thus, the overall impacts of running on V1 units again appear large and positive in mice, and much smaller (and perhaps slightly negative) in marmoset.

Given these apparently categorical differences between the two species at the levels of both experimental sessions (*Figure 2*) and individual units (*Figure 3*), a key question is whether mouse and marmoset visual cortices are modulated by non-visual input in fundamentally different ways. To answer this, we employed more powerful model-based neuronal population analyses that inferred trial-to-trial variations in shared gain modulations across V1 (*Figure 4a and d*; *Whiteway et al., 2019*), in a manner totally agnostic to running (or any other aspect of behavior). This shared-gain model improved descriptions of the population data over simpler models that only took the stimulus (and slow drifts in baseline firing rate) into account for all sessions (*Figure 4b and c*; marmoset p=$1.52 \times 10^{-82}$, stat = 27174, n = 754, Wilcoxon signed-rank test; mouse p=$4.64 \times 10^{-181}$, stat = 25966, n = 1257). This was true in both species, bolstering the emerging notion that population-level gain modulations are a general principle of mammalian V1 function (*Whiteway et al., 2019*; *Lin et al., 2015*; *Arandia-Romero et al., 2016*; *Goris et al., 2014*; *Ferguson and Cardin, 2020*). This shared gain term modulated more strongly in mice compared to marmosets (*Figure 4e*, std. dev. in mouse = 2.170 [2.106, 2.245], marmoset = 1.188 [1.072, 1.274], p<$1 \times 10-9$, stat = 1013202, Mann×Whitney *U* test). Furthermore, in the mouse, shared gain was higher for running than stationary as estimated during stimulus presentations (mean difference 0.970 [0.761, 1.225], p~0, stat 8.017, *t*-test), demonstrating that a substantial portion of modulations of mouse V1 can be explained by a shared gain term that increases with running (*Figure 4f*, orange point). In marmoset, shared gain was slightly but reliably lower when running (mean difference = −0.125 [-0.203, -0.059], p=0.002, stat = −3.360, *t*-test, *Figure 4f*, blue point), a quantitatively very different relation to running than in mouse (p=$8.77 \times 10^{-9}$, stat = 6.615, two-sample *t*-test). Thus, a common mechanism (shared gain) can describe running modulations in both species, but with quantitatively different correlations with behavior that make for potentially distinct downstream impacts on perception and action.

Although our marmoset dataset focused on V1 neurons representing the central portion of the visual field, we were also able to record simultaneously from neurons with peripheral and central

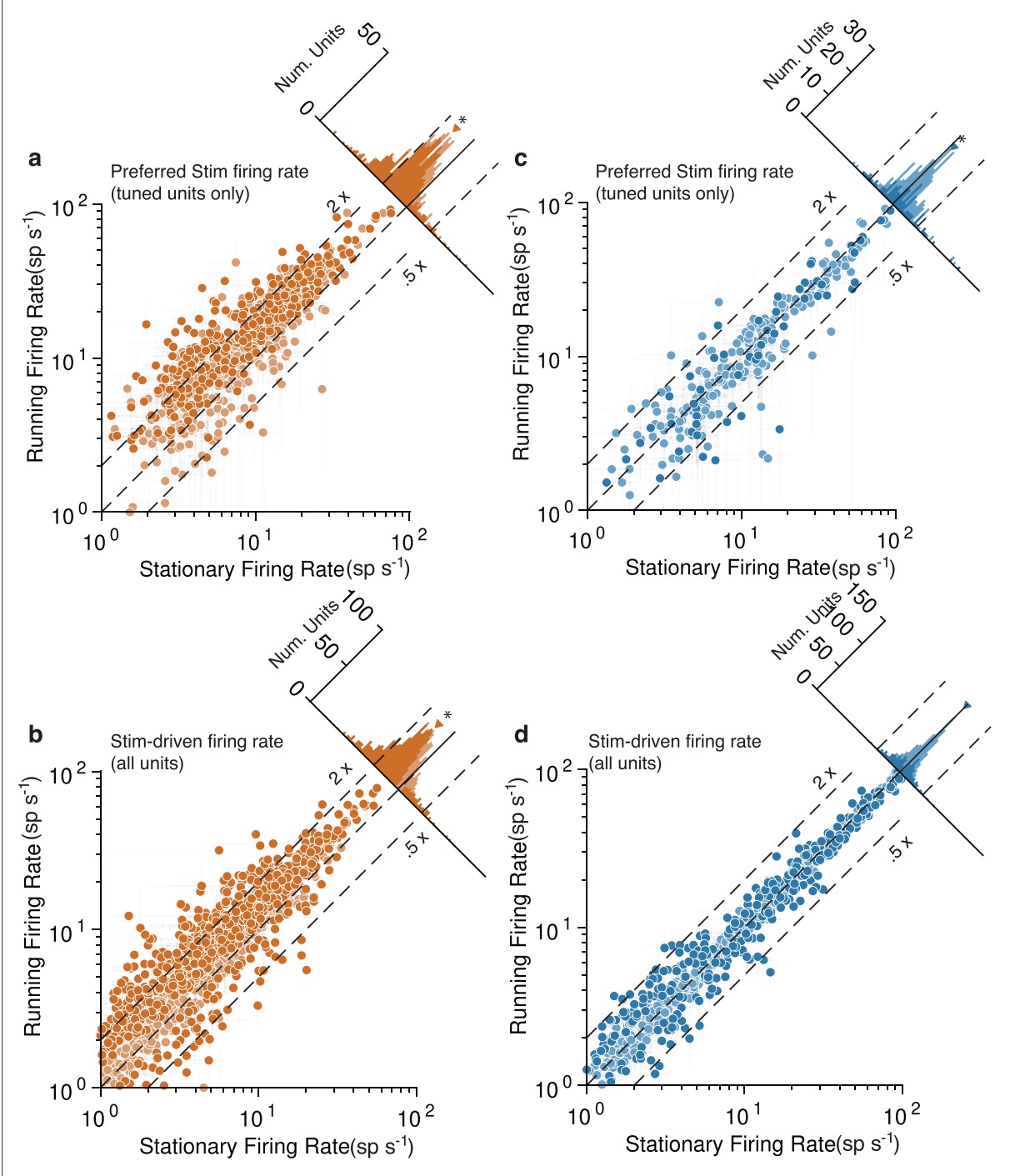

**Figure 3.** Running strongly increases mouse V1 activity and subtly decreases marmoset V1 activity, evidenced at the level of individual units. Mouse data points are plotted in orange and marmoset data in blue. (**a**) Scatterplot (log-log) shows firing rate to preferred stimulus for tuned units (orientation-selectivity indices [OSI] > 0.2), during running (y-axis) and stationary (x-axis). Histogram summarizes the projections onto the line of unity and shows a clear shift indicating increases in response during running (geometric mean ratio [running/stationary] = 1.523 [1.469, 1.579], n = 743). Dark-shaded symbols indicate individually significant units. Dashed lines indicate doubling (2×) and halving (0.5×) of response. (**b**) Same format, but now showing the response aggregated over all stimuli, for all units (geometric mean ratio [running/stationary] = 1.402 [1.365, 1.440], n = 1168). A similar pattern reflecting primarily large increases is evident. (**c, d**) V1 units in marmoset show a very different pattern. Responses of tuned units to preferred stimuli (**c**) cluster more closely to the line of unity, with a small but significant shift indicating a subtle decrease in response (geometric mean ratio [running/stationary] = 0.899 [0.851, 0.949], n = 228). Responses to all stimuli for all units (**d**) show even less running-related modulation (geometric mean ratio [running/stationary] = 1.011 [0.995, 1.027], n = 786).

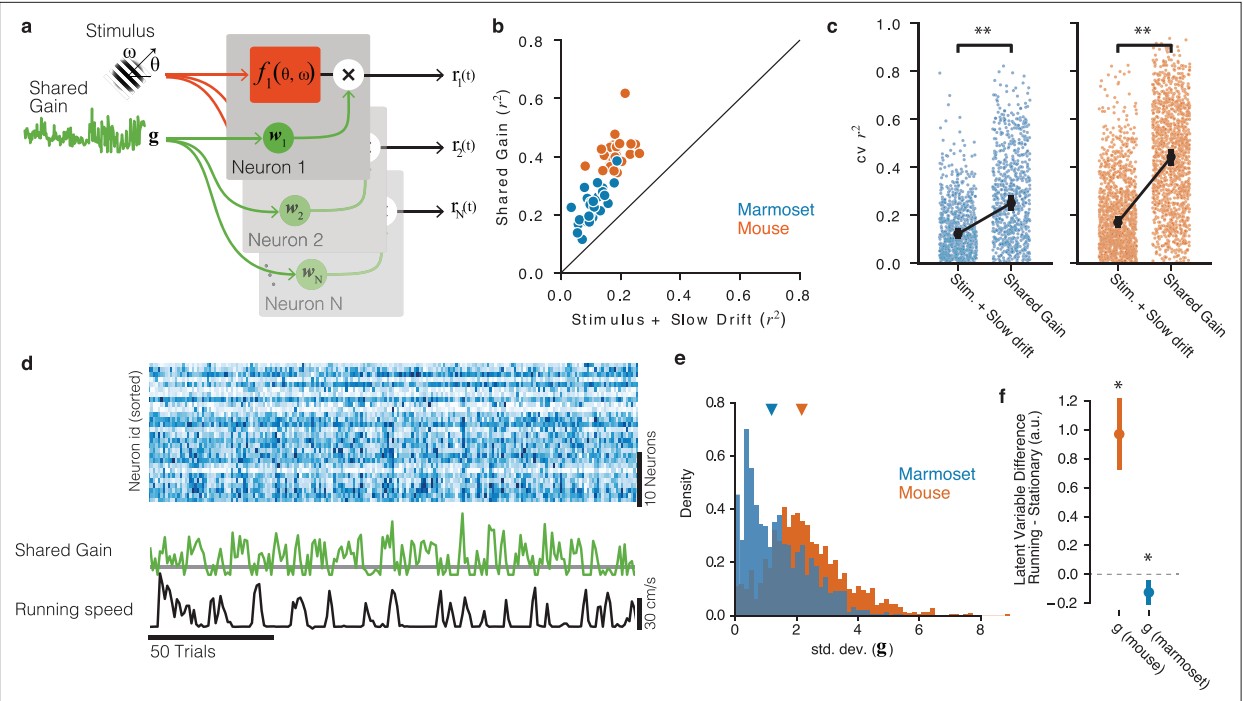

**Figure 4.** Shared gain model accounts for fluctuations in both mouse and marmoset V1, and explains species differences. (**a**) Structure of shared modulator model. In addition to the effects of the stimulus (and slow drift in responsiveness, not rendered), the model allows for a shared gain/multiplicative term (green). Each simultaneously recorded neuron is fitted with a weight to the latent gain term. (**b**) The resulting model provides a better account of both mouse and marmoset V1 responses compared to a simple model that only fits stimulus and slow drift terms. Points show variance explained ($r^2$) on test data for each session under each of the two models, plotted against one another. (**c**) Variance explained for individual units was significantly improved in both species (marmoset: gain model [median $r^2$ = 0.2504] significantly higher than stim + drift [median $r^2$ = 0.1220], p=$1.52 \times 10^{-82}$, stat = 27174, Wilcoxon signed-rank test; mouse: gain model [median $r^2$ = 0.4420] significantly better than stim + drift [median $r^2$ = 0.1697], p=$4.64 \times 10^{-181}$, stat = 25966, Wilcoxon signed-rank test). (**d**) Example of relationship between neural responses (top raster, blue), the shared gain (green), and running speed (black trace). Visual inspection similar to that in *Figure 2* can be performed. (**e**) Gain modulations span a larger range in mice than in marmosets. Orange, gain term from each mouse session; blue, gain term from each marmoset session. Triangles indicate medians (mouse = 2.17 [2.11, 2.25], marmoset = 1.19 [1.07, 1.27]). (**f**) Shared gain term is larger during running for mouse data, but is slightly smaller during running for marmoset data (difference is plotted on y-axis; mouse = 0.970 [0.761, 1.225], p=$4.73 \times 10^{-9}$, stat 8.017, one-sample *t*-test; marmoset = −0.125 [−0.203, −0.059], p=0.002, stat = −3.360, one-sample *t*-test).

(foveal) receptive fields by advancing a Neuropixels probe into both the superficial portion of V1 (foveal/central) and the calcarine sulcus (peripheral), resulting in simultaneous recordings of 110 and 147 (stimulus-driven) units representing the central and peripheral portions of the visual field, respectively. Analyzing neurons with peripheral receptive fields separately revealed a difference in running modulations between these retinotopically distinct portions of V1: peripheral neurons had slightly higher stimulus-driven responses during running (aggregating over all stimuli, geometric mean ratio [running/stationary] = 1.129 [1.068, 1.194], n = 147; difference with the central units was significant, p=2.100e-03, stat = 12376, Mann–Whitney *U* test), and the two sessions in which we were able to perform these measurements had higher positive correlations than any sessions in our entire foveal V1 dataset (assessed by correlating running speed either with the First Neural PC or with a shared gain term). Although the foveal representation in V1 (accessible in marmosets on the dorsal surface of the brain) is slightly suppressed by running, it appears that quantitative differences exist in the peripheral representation (which we recorded from in the calcarine sulcus). This initial set of recordings suggests that subtle increases in response might occur in the peripheral representation in marmoset V1. This finding calls for a larger-scale study of how such modulations might differ across portions of the retinotopic map, and for further consideration of the implications for cross-species comparisons. An intriguing conjecture is that the primate foveal representation might be functionally unique, but that the primate peripheral representation might be more functionally similar to that of mouse V1 (*Horrocks et al., 2022*).

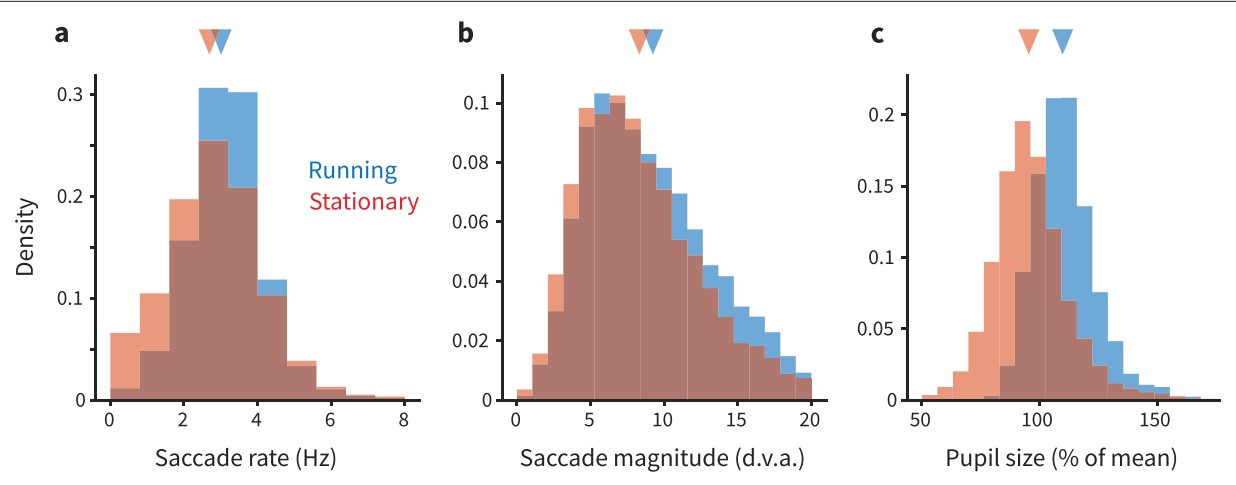

**Figure 5.** Eye movements and pupil size are modestly different during running. (**a**) Each panel shows overlapping histograms of a measurement made on trials when the animal was running (blue) or stationary (red). (**a**) Saccade rate (in Hz) is slightly higher during running. (**b**) Saccade magnitude (in degrees of visual angle) is also slightly higher during running. The slight differences in saccade frequency and size did not quantitatively explain the differences in neural activity during running versus stationary periods; see main text for analysis. (**c**) Pupil size (expressed as % of mean size during stationary) is 8% larger during running (see main text for additional quantification).

Although this is an interesting potential distinction that further work will investigate more systematically, we emphasize that the main result described earlier still holds: the effects of running are small in marmoset V1. Even though there are slightly positive modulations in the peripheral representation (and hence, are of the same sign as those in mouse), the magnitude of whatever running-correlated modulations we could measure in marmoset V1 are still small relative to those in mouse V1 (median spike rate modulation by running significantly different between mouse and marmoset calcarine/peripheral recordings: p=$7.639 \times 10^{-11}$, stat = 7967825, Mann–Whitney $U$ test).

Finally, we assessed whether the modest running-correlated modulations we observed in marmoset V1 might be explained by eye movements. If eye movements differed when the animal ran versus when it did not run, that would mean that the retinal input differed between the two conditions (*Horrocks et al., 2022*). In that case, running modulations would not reflect a direct effect of running (a fundamentally non-visual effect), but rather a consequence of changes in the patterns of retinal stimulation (which we already know affects V1 responses). To test this possibility, we quantified the number of saccades per stimulus presentation, as well as saccade size (vector magnitude), and then assessed whether these eye movement metrics differed as a function of running.

We found that eye movements were quite similar between running and stationary periods, although subtle quantitative differences were revealed (*Figure 5*). In short, saccades were slightly more frequent and larger during running (saccade frequency during running: 2.653 Hz, 95% CI [2.600, 2.697]; stationary/not running: 2.525 Hz [2.475, 2.573]; saccade magnitude during running: 9.261° [9.140, 9.374]; stationary/not running: 8.337° [8.190, 8.470]). This result motivated us to then assess whether these running-correlated eye movement differences might quantitatively explain the running-correlated modulations of V1 response. Our initial analyses found that differences in retinal stimulation due to differences in eye movements are unlikely to explain running-correlated suppression of V1 activity. We used linear regression to estimate the relationship between number of saccades and the firing rate of each unit in each trial. This enabled us to predict how much change in firing rate we should expect given the differences in saccade rate between the running and stationary conditions. The response change predicted from saccades was much less than the already-small running-correlated changes in response we observed in our experiment. On the aggregate, saccades slightly increased activity (predicted spike rate increase during running based on saccades = 0.05 Hz; expressed as gain, <1%), and thus cannot explain the sign or magnitude of the subtle decreases we observed. In short, the decreases in V1 activity we saw in our main dataset are not likely to be explained by differential patterns of eye movements. (Likewise, the distinction we saw between running-correlated modulations in foveal versus peripheral V1 is unlikely attributable to eye movements, as we recorded

simultaneously from both parts of the retinotopic map, meaning that the eye movements were the same despite the difference in modulations.)

Regrettably, the mouse dataset with which we compared our marmoset recordings did not reliably have eye video with quality required to do precise gaze estimation (at least for many of the sessions), so we could not perform a definitive analysis in mouse. However, the degree to which movement-correlated modulations of sensory processing contain a retinal contribution is an important issue (*Horrocks et al., 2022*), and one that we hope to tackle more directly in future cross-species studies wherein eyetracking and knowledge of individual receptive fields is highly, and equally, prioritized in mice and marmosets.

We also analyzed the pupil size from our eyetracking videos. Pupil size was ~8% larger during running. This finding is consistent with the idea that the marmosets were in a higher arousal state during running. Such a result is at least loosely consistent with effects seen in mice (although in that literature, there is some degree of dissociation between modulations due to arousal and those due to running per se; *Vinck et al., 2015*). Thus, the changes in pupil size we detected do suggest that when a marmoset runs, it is likely in a higher arousal state, similar to that in mice. However, more work would be required to perform cross-species calibrations to understand how the magnitude of changes in pupil size corresponds to changes in levels of arousal. At this point, we can conclude that the differences we see in the size (and sometimes, sign) of V1 modulations across species are unlikely due to a categorical difference in a link between running and arousal in mice versus marmosets, but possible quantitative distinctions deserve further consideration.

## Discussion

In short, running does not affect V1 activity in marmosets like it does in mouse. The large, typically positive correlations between running and V1 activity often found in mice are simply not evident in marmosets. Although we matched our experimental protocol to mouse experiments and used the same metrics and analysis pipeline, the difference in results across species was stark. We hypothesize that this distinction holds at the level of taxonomic order, distinguishing how much behavioral state interacts with early stages of visual processing in primates versus rodents.

Diving deeper into the pattern of results, we did detect small (but statistically nonzero) modulations of marmoset V1 response correlated with running. In the foveal representation in V1 – where we made the majority of our recordings – responses on average were slightly smaller during running; in the peripheral representation, responses were slightly larger.

Despite the main result of this study being that running-correlated modulations in marmoset V1 are small, and hence quantitatively different than that in mouse V1, our population-level analyses did point towards a possible cross-species generalization. The same shared-gain model improved accounts of both mouse and marmoset V1 activity. These population-level gain modulations likely reflect modulatory inputs associated with behavioral state and arousal. This commonality connects with mechanistic knowledge of how V1 activity is modulated. The primate-rodent difference in the magnitude and sign of V1 gain modulations we observed is in fact consistent with known differences in neuromodulatory inputs related to arousal in rodent and primate V1 (*Disney and Robert, 2019*; *Coppola and Disney, 2018*). In primates, the locations of ACh receptors allow cholinergic inputs to increase the activity of the majority of GABAergic neurons and hence suppress net activity via inhibition (*Disney et al., 2007*; *Lien and Scanziani, 2013*), but pharmacologically and anatomically distinct cholinergic influences in rodent likely exert more complex effects on net activity, including disinhibition which can increase net activity (*Pakan et al., 2016*; *Fu et al., 2014*; *Pfeffer et al., 2013*). Our population-level analyses also lay groundwork for connections to indirect and aggregate measures of neural activity made in humans under related conditions (*Chen et al., 2022*; *Cao et al., 2020*; *Benjamin et al., 2018*), as well as the typically small modulations seen in primate visual cortices elicited by carefully controlled attentional tasks, which are more clear when population-level modulations are considered (*Mitchell et al., 2009*; *Cohen and Maunsell, 2009*; *Rabinowitz et al., 2015*).

We also performed an analysis of whether eye movements might contribute to differential visual (retinal) stimulation, which in turn could differentially modulate visually driven activity in V1 during running versus stationary periods. We found that there were subtle increases in eye movement frequency and saccade amplitude during running. However, saccades on average slightly increased V1 activity, so it seems unlikely that eye-movement-mediated changes in retinal stimulation explain

the modest decreases we observed during running. We found that analysis of eye movements was difficult in some of the mouse datasets. Because receptive fields in mouse V1 can be very large, uncontrolled (and/or uncharacterized) eye movements can not only create visual modulations of the stimulus on the screen, but can also hit the edges of the monitor under some viewing conditions. A related study *Talluri et al., 2023* found that eye movements (or, their effects on retinal stimulation) explained all of the modulations of V1 activity that were correlated with facial/body movements in seated macaques. Further work will be needed to understand how much eye movements play a role in both running-correlated and movement-correlated modulations in the mouse. This will require monitoring eye movements and dissecting the ensuing retinal effects from those of other (body and face) movements (*Musall et al., 2019*); all of these types of motor activity (and subsequent 'sensory reafference') may be partially correlated.

Our results (as well as those of *Talluri et al., 2023*) reveal a number of additional issues that should be addressed in follow-up work to even more tightly relate work across the two species. In our study, we attempted to match the overall treadmill apparatus and the visual stimuli used in the mouse studies. Even that required species-specific customization of the treadmill, as well as taking into account the higher spatial acuity of primate vision (which is why our study used much higher spatial frequencies in our set of drifting gratings). We describe how additional unresolved issues could be addressed for improved cross-species integration.

First, we analyzed pupil size and found that it was larger when the marmosets ran. At first glance, this suggests that running does indicate a more aroused internal state in the marmosets, as it likely does in mice (*Vinck et al., 2015*). However, it is less clear whether the magnitude of pupil size changes in marmosets corresponds to the same amount of arousal change that occurs in mice. Relative calibration of the dynamic range of pupil size (and measuring other biomarkers of arousal) may make for more satisfying inferences about internal states across species, as it has been shown that some (but not all) of running-correlated modulations are likely due to arousal (*Vinck et al., 2015*; *Lee et al., 2014*).

Second, although we found only small effects (relative to mouse) at the aggregate level, our results call for more specific investigations of modulations at the level of cell types and subcircuits (*Niell and Stryker, 2010*; *Pakan et al., 2016*; *Bennett et al., 2013*; *Polack et al., 2013*). Such investigations may reveal more nuanced effects in primate V1, using tools that can better unpack the circuitry associated with factors such as cholinergic modulation, which are known to differ in important ways across rodent and monkey (*Coppola and Disney, 2018*; *Disney and Robert, 2019*). Additionally, differences in feedback circuits also exist across the visual field representation within primate V1 (*Wang et al., 2022*). This – and the proposition that mouse V1 may be a better model of primate peripheral vision (*Horrocks et al., 2022*) – has motivated us to perform more systematic and larger-scale recordings to compare the foveal and peripheral representations.

Third, our results call for additional study across other visual areas. In mice, the large effects on V1 activity are likely to affect all subsequent stages of processing (*Christensen and Pillow, 2022*), but in marmosets, the small effects are less likely to have pronounced downstream effects. That said, running may directly and more strongly interact with later stages of visual processing in primates. This would be consistent with differences in where canonical computations occur across species with different numbers of visual areas (*Felleman and Van Essen, 1991*; *Garrett et al., 2014*; *Scholl et al., 2013*). Such measurements in primate extrastriate visual areas are already in progress in our laboratory.

Finally, larger effects of behavioral state may still be found in primate V1: other behaviors that more directly recruit active vision may reveal stronger modulations. In mice, running may have a more direct functional relation to visual processing. Marmosets may instead wish to recruit head or body movements that are not realizable in the head-fixed preparation that we used for eye-movement and neural recording. These questions will be addressed in freely moving and head-free subjects.

Although our main result is simply that running-correlated modulations in marmoset V1 are small relative to those in mouse, we did find evidence for behaviorally correlated population-level gain modulations in both species. This sort of commonality may support further cross-species generalizations that transcend simpler observations of empirical similarity or dissimilarity (*Niell and Scanziani, 2021*; *Priebe and McGee, 2014*). Further work explicating how shared basic mechanisms may ultimately result in rather different patterns of interaction between vision and action will be critical for linking our understanding of cortical function between currently preferred model organisms and

across taxonomic orders. The results in this report reflect just the starting point for a larger comparative inquiry.

## Materials and methods

We performed electrophysiological recordings in V1 of two common marmosets (one male, 'marmoset G', and one female, 'marmoset B', both aged 2 years). Both subjects had chronically implanted N-form arrays (Modular Bionics, Modular Bionics, Berkeley, CA) inserted into left V1. Implantations were performed with standard surgical procedures for chronically implanted arrays in primates. Additional recordings were also performed using Neuropixels 1.0 probes (*Jun et al., 2017*) acutely inserted into small craniotomies (procedure described below). All experimental protocols were approved by The University of Texas Institutional Animal Care and Use Committee and in accordance with National Institute of Health standards for care and use of laboratory animals.

Subjects perched quadrupedally on a 12″ diameter wheel while head-fixed facing a 24″ LCD (BenQ) monitor (resolution = 1920 × 1080 pixels, refresh rate = 120 Hz) corrected to have a linear gamma function, at a distance of 36 cm (pixels per degree = 26.03) in a dark room. Eye position was recorded via an Eyelink 1000 eye tracker (SR Research) sampling at 1 kHz. A syringe pump-operated reward line was used to deliver liquid reward to the subject. Timing events were generated using a Datapixx I/O box (VPixx) for precise temporal registration. All of these systems were integrated in and controlled by MarmoView. Stimuli were generated using MarmoView, custom code based on the PLDAPS (*Eastman and Huk, 2012*) system using Psychophysics Toolbox (*Brainard, 1997*) in MATLAB (MathWorks). For the electrophysiology data gathered from the N-Form arrays, neural responses were recorded using two Intan C3324 headstages attached to the array connectors which sent output to an Open Ephys acquisition board and GUI on a dedicated computer. In electrophysiology data gathered using Neuropixels probes, data was sent through Neuropixels headstages to a Neuropixels PXIe acquisition card within a PXIe chassis (National Instruments). The PXIe chassis sent outputs to a dedicated computer running Open Ephys with an Open Ephys acquisition board additionally attached to record timing events sent from the Datapixx I/O box. Spike sorting on data acquired using N-Form arrays was performed using in-house code to track and merge data from identified single units across multiple recording sessions (*Muthmann et al., 2021*). Spike sorting for data acquired using Neuropixels probes was performed using Kilosort 2.5.

### Chronic N-Form array recordings

Chronic array recordings were performed using 64-channel chronically implanted 3D N-Form arrays consisting of 16 shanks arrayed in a 4 × 4 grid with shanks evenly spaced 0.4 mm apart (Modular Bionics, Berkeley). Iridium oxide electrodes are located at 1, 1.125, 1.25, and 1.5 mm (tip) along each shank, forming a 4 × 4 × 4 grid of electrodes. Arrays were chronically inserted into the left dorsal V1 of marmosets G and B at 1.5 and 4° eccentric in the visual field, respectively (confirmed via post hoc spatial RF mapping). Well-isolated single units were detectable on the arrays in excess of 6 months after the initial implantation procedure.

### Acute Neuropixels recordings

Acute Neuropixels recordings were performed using standard Neuropixels 1.0 electrodes (IMEC, Leuven, Belgium). Each probe consists of 384 recording channels that can individually be configured to record signals from 960 selectable sites along a 10 mm long, 70 × 24 µm cross-sectional straight shank. Probes were lowered into right dorsal V1 of marmoset G via one of three burr holes spaced irregularly along the AP axis 4–5 mm from the midline for a single session of experiments. Natural images were played to provide visual stimulus as well as occupy the subject and keep them awake during insertion and probe settling. The temporary seal on the burr hole was removed, the intact dura nicked with a thin needle and the burr hole filled with saline. The probe was then lowered through the dural slit at 500 µm/min, allowing 5 min for settling every 1000 µm of total insertion. The whole-probe LFP visualization was monitored during insertion for the characteristic banding of increased LFP amplitude that characterizes cortical tissue. The probe was inserted until this banding was visible on the electrodes nearest the tip of the probe, indicating that the probe tip itself had passed through the dorsal cortex and was within the white matter. The probe was then advanced until a second band

became visible on the electrodes nearest the tip, indicating the tip of the probe had exited through the cortex of the calcarine sulcus. The probe was then advanced slightly until the entirety of the second LFP band was visible to ensure that electrodes covered the full depth of the calcarine cortex and the tip of the probe was located confidently within the CSF of the sulcus. The probe was then allowed to settle for 10 min. Active electrode sites on the probe were configured to subtend both dorsal and calcarine cortex simultaneously. Post hoc receptive field recreation confirmed that visually driven, tuned, V1 neurons were recorded at both foveal and peripheral eccentricities.

## Mouse dataset from Allen Institute

Mouse data were downloaded from the publicly available Visual Coding database at https://portal. brain-map.org/explore/circuits/visual-coding-neuropixels. We used the same analysis code to analyze these data and the marmoset data we collected.

## General experimental procedure

Marmoset recording sessions began with eyetracking calibration. Once calibration was completed, the wheel was unlocked and the subject was allowed to locomote freely, head-fixed, while free-viewing stimuli. Trials for all stimuli were 20 s long with a 500 ms ITI and a 20-s-long natural image interleaved every fifth trial to keep the subject engaged. Stimuli were shown in blocks of 10 min and a typical recording session consisted of 50 trials of calibration followed by one or two blocks of a drifting grating stimulus and one block each of the two mapping stimuli. To elicit sufficiently reliable and frequent running behavior, subjects were rewarded at set locomotion distance intervals unrelated to the stimulus or gaze behavior (typical rewards were 50–70 μL and distance required to achieve a reward usually varied between 20 and 75 cm; reward amounts and intervals were adjusted daily to maximally motivate the subject).

## Eyetracking calibration

While the wheel was locked, subjects were allowed to free-view a sequence of patterns of marmoset faces. Marmosets naturally direct their gaze towards the faces of other marmosets when allowed to free-view with little-to-no training, allowing for the experimenter to adjust the calibration offset and gain manually between pattern presentations. Faces were 1.5° in diameter and were presented for 3 s with a 2 s ISI between patterns. A portion of presented patterns were asymmetrical across both the X and Y axes of the screen to allow for disambiguation in the case of axis sign flips in the calibration. Fifty trials were presented before each recording session to verify and refine the calibration. Calibration drift between sessions was minimal, requiring minor (<1°) adjustments over the course of 1–2 months of recordings.

## Drifting grating stimuli

The primary stimulus consisted of full-field drifting gratings. Gratings were optimized to drive marmoset V1 with 3 separate spatial frequencies (one, two, and four cycles per degree), 2 drift speeds (1 or 2° per second), and 12 orientations (evenly spaced 30° intervals). Each trial consisted of multiple grating presentations, each with a randomized spatial frequency, drift speed, and orientation. Gratings were displayed for 833 ms followed by a 249–415 ms randomly jittered inter-stimulus interval. After each 20 s trial, there was a longer 500 ms inter-trial interval. Every fifth trial was replaced with a natural image to keep subjects engaged and allow for visual assessment of calibration stability on the experimenter's display.

## Mapping of receptive fields

A spatiotemporal receptive field mapping stimulus, consisting of sparse dot noise, was shown during each recording session. One hundred 1° white and black dots were presented at 50% contrast at random points on the screen. Dots had a lifetime of two frames (16.666 ms). Marmosets freely viewed the stimulus, and we corrected for eye position offline to estimate the spatial receptive fields using forward correlation (*Yates et al., 2021*).

## Necessary differences between mouse and marmoset experiments

Although we sought to perform experiments in marmosets that were as similar as possible to mouse experiments, some differences in their visual systems and behavior made for differences. Because the

spatial frequency tunings of marmoset and mouse V1 neurons are starkly different, we used stimuli with considerably higher spatial frequencies than in the mouse experiments. Relatedly, marmoset V1 receptive fields are much smaller than in mouse. Because we used full-field stimuli (to match mouse experiments), responses in marmoset V1 were likely affected by substantial amounts of surround suppression, which would reduce overall responses. We also learned that, although the marmosets were comfortable perched on the wheel treadmill, they did not naturally run enough for our experimental purposes. We therefore incorporated a reward scheme to motivate the subjects to run more frequently. Finally, the mouse dataset we analyzed comprised a large number of mice with a small number of sessions per mouse; as is required of work with nonhuman primates, we were limited to a smaller number of subjects (N = 2) and ran many experimental sessions with each animal.

### Session and cell inclusion criteria

For the analyses shown in *Figure 2*, sessions were included if they contained more than 250 trials and a proportion of trials running was not less than 10% or greater than 90%. For the mouse dataset, this yielded 25/32 sessions. For the marmoset dataset, this yielded 27/34 sessions. For the unit-wise analyses in *Figure 3*, super-sessioned units were included for analysis if they had more than 300 trials of data and a mean firing rate of >1 spike/s. This yielded 1168/2015 units in mouse and 786/1837 units in marmoset.

For the analyses shown in *Figure 4*, sessions were included using the same trial and running criterion as in *Figure 2*. Only units that were well fit by the stimulus + slow drift model (i.e., cross-validated better than the null, see 'Shared modulator model') were included and sessions were excluded if fewer than 10 units met this criterion. This resulted in 31/32 sessions for mouse and 28/34 sessions for marmoset.

### Analysis of tuning

We counted spikes between the 50 ms after grating onset and 50 ms after grating offset and divided by the interval to generate a trial spike rate. To calculate orientation tuning curves, we computed the mean firing rate of each orientation and spatial frequency. Because we were limited by the animal's behavior to determine the number of trials in each condition (i.e., running or not), we computed orientation tuning as a weighted average across spatial frequencies with weights set by the spatial frequency tuning. We used these resulting curves for all analyses of tuning. We confirmed that the results did not change qualitatively if we either used only the best spatial frequency or marginalized across spatial frequency.

Orientation selectivity index was calculated using the following equation:

$$\text{OSI} = \frac{\sqrt{[r^T \sin(2\theta)]^2 + [r^T \cos(2\theta)]^2}}{\sum(r)}$$

where $\theta$ is the orientation and $r$ is the baseline-subtracted vector of rates across orientations.

### Analysis of eye movement effects on neural response

To assess whether and how eye movements might differ between running and stationary periods (and perhaps account for some or all of the running-correlated modulations of V1 response), we started by counting the number of saccades within a bin corresponding to each stimulus presentation (from 0.2 s before stimulus onset to 0.1 after offset), as well as calculating the average saccade size (vector magnitude) of those saccades. We then regressed these terms against the spike count in each bin, allowing us to estimate the effect of eye movements in units of spike rate (Hz). (We also analyzed the variance of the eye position signal and got similar results.) For the analysis of pupil size, we used the values returned by our Eyelink eyetracker, averaged in the same bins as for the saccade analyses.

### Shared modulator model

To capture shared modulator signals in an unsupervised manner, we fit our neural populations with a latent variable model (*Whiteway and Butts, 2019*). The goal of our latent variable model was to summarize population activity with low-dimensional shared signal that operates as a gain on the stimulus processing (e.g., *Goris et al., 2014*; *Lin et al., 2015*). In this model, the response of an individual neuron, $\mathbf{r}_i$ on trial $t$ is given by

$$\mathbf{r}_i(t) = f_i[s(t)]g_i(t) + b_i \tag{1}$$

where the stimulus response $f_i[s(t)]$ is given by the tuning curve, $g_i(t)$ is a neuron-specific gain on the stimulus response, and $b_i$ is the baseline firing rate. Similar models have been employed to describe the population response in V1 in several species (*Goris et al., 2014*; *Lin et al., 2015*; *Arandia-Romero et al., 2016*; *Whiteway et al., 2019*).

Because the gain signal is shared across neurons, we fit this model to all $n$ neurons in a given recording at the same time. To capture the stimulus tuning curves, we represented the stimulus on each trial $s(t)$ as an $m$-dimensional 'one-hot' vector, where $m$ is the number of possible conditions (Orientation × Spatial Frequency) and on each trial all elements are zero, except for the condition shown. Thus, $f[s(t)]$ becomes a linear projection of the stimulus on the tuning curves, $\mathbf{A}s(t)$, where $\mathbf{A}$ is an $n \times m$ matrix of tuning weights. We decomposed the gain for each neuron on each trial into a rank 1 matrix that was rectified and offset by one, $g(t) = \text{ReLU}[1 + z(t)\mathbf{w}]$, where $w$ is an $n$-dimensional vector of loadings that map the one-dimensional trial latent $z(t)$ to a population-level signal, $z(t)\mathbf{w}$. This signal is offset by 1 and rectified such that it is always positive and a loading weight of zero equals a gain of 1.0.

To capture any unit-specific slow drifts in firing rate, we further parameterized $\mathbf{b}$ as a linear combination of five b0-splines evenly spaced across the experiment (*Quinn et al., 2021*). Thus, the baseline firing rate for each neuron, $i$, was a linear combination of five 'tent' basis functions spaced evenly across the experiment, $\mathbf{b}_i(t) = \sum_j b_j \phi_j(t)$.

Thus, the full model describes the population response as

$$\mathbf{r}(t) = \mathbf{A}s(t)\text{ReLU}[1 + z(t)\mathbf{w}] + \mathbf{b}(t) \tag{2}$$

The parameters of the model are the stimulus tuning parameters $\mathbf{A}$, the shared gain, $z$, the gain loadings, $\mathbf{w}$, and the 'tent' basis weights, $b_{i,j}$'s.

We first fit a baseline model with only stimulus and baseline parameters

$$\mathbf{r}(t) = \mathbf{A}s(t) + \mathbf{b}(t) \tag{3}$$

Following *Whiteway and Butts, 2017*, we initialized $\mathbf{A}$ and $\mathbf{b}$ using fits from a model without latent variables and initialized the latent variable, $z$, and loadings, $\mathbf{w}$, using an Autoencoder (*Bengio et al., 2013*; *Whiteway and Butts, 2017*). We then fit the gain, loadings, and stimulus parameters using iterative optimization with L-BFGS, by minimizing the mean squared error (MSE) between the observed spikes and the model rates. The model parameters were regularized with a modest amount of L2-penalty and the amount was set using cross-validation on the training set. The latent variables were penalized with a small squared derivative penalty to impose some smoothness across trials. This was set to be small and the same value across all sessions. We reverted the model to the autoencoder initialization if the MSE on a validation set did not improve during fitting.

We cross-validated the model using a speckled holdout pattern (*Williams et al., 2018*) whereby some fraction of neurons were withheld on each trial with probability p=0.25. We further divided the withheld data into a validation set and a test set by randomly assigning units to either group on each trial with probability 0.5. The validation loss was used to stop the optimization during the iterative fitting and the test set was used to evaluate the models.

## Sign of latent variables

We anchored the sign of all latent variables to the average firing rate of the population, such that positive means increases in the average firing rate. In general, for both the principal components analysis (*Figure 2*) and the shared population modulation model (*Figure 4*), the sign of the latent variable is arbitrary and only becomes a signed effect once multiplied by the loading weight for each unit. Thus, to interpret these values, we used the loadings to flip the sign of the latent to be positive for increases in the average firing rate. Specifically, we took the largest eigenvector, $u$, of the covariance matrix across neurons and modified the sign such that the average sign was positive: $\mathbf{u} = \mathbf{u} \times \text{sign}\left(\sum_i \text{sign}(\mathbf{u}(i))\right)$, where $sign(x)$ is 1 for $x \geq 0$ and –1 for $x < 0$. We then projected the

mean-subtracted firing rates on (the sign-corrected) $u$. This gives a '1st PC' with an interpretable sign. For analyses of shared gain (*Figure 4*), we projected back to the population space and then averaged the per-unit gain.

## Acknowledgements

We thank Allison Laudano for animal and colony management and care, Christopher Badillo for apparatus design and fabrication, and Nika Hazen for assistance with animal work. Cris Niell, Cory Miller, Jude Mitchell, and Anne Churchland all provided valuable feedback on drafts of this paper. We thank the Visual Coding team at the Allen Institute for sharing the mouse data used in this paper (https://portal.brain-map.org/explore/circuits/visual-coding-neuropixels).

## Additional information

### Funding

| Funder | Grant reference number | Author |
|---|---|---|
| National Institutes of Health | BRAIN Initiative U01-UF1NS116377 | Alexander C Huk |
| Air Force Office of Scientific Research | FA9550-19-1-0357 | Alexander C Huk |
| National Science Foundation | NSC-FO 2123605 | Daniel A Butts |
| National Institutes of Health | K99EY032179-02 | Jacob Yates |

The funders had no role in study design, data collection and interpretation, or the decision to submit the work for publication.

### Author contributions

John P Liska, Conceptualization, Data curation, Software, Investigation, Methodology, Writing – original draft, Writing – review and editing; Declan P Rowley, Data curation, Software, Formal analysis, Validation, Investigation, Visualization, Methodology, Writing – review and editing; Trevor Thai Kim Nguyen, Jens-Oliver Muthmann, Methodology, Writing – review and editing; Daniel A Butts, Conceptualization, Formal analysis, Supervision, Funding acquisition, Methodology, Writing – review and editing; Jacob Yates, Conceptualization, Data curation, Software, Formal analysis, Supervision, Funding acquisition, Validation, Visualization, Methodology, Writing – original draft, Writing – review and editing; Alexander C Huk, Conceptualization, Supervision, Funding acquisition, Methodology, Writing – original draft, Project administration, Writing – review and editing

### Author ORCIDs

Trevor Thai Kim Nguyen ⓘ https://orcid.org/0009-0005-3277-9535
Jens-Oliver Muthmann ⓘ https://orcid.org/0000-0002-0439-9704
Daniel A Butts ⓘ https://orcid.org/0000-0002-0158-5317
Jacob Yates ⓘ https://orcid.org/0000-0001-8322-5982
Alexander C Huk ⓘ https://orcid.org/0000-0003-1430-6935

### Ethics

This study was performed in strict accordance with the recommendations in the Guide for the Care and Use of Laboratory Animals of the National Institutes of Health. All of the animals were handled according to approved institutional animal care and use committee (IACUC) protocols at The University of Texas at Austin and the University of California, Los Angeles (IACUC approval: UCLA, ARC-2022-085). All surgery was performed under anesthesia, and every effort was made to minimize suffering.

Reviewer #1 (Public review): https://doi.org/10.7554/eLife.87736.3.sa1

Reviewer #2 (Public review): https://doi.org/10.7554/eLife.87736.3.sa2
Reviewer #3 (Public review): https://doi.org/10.7554/eLife.87736.3.sa3
Author response https://doi.org/10.7554/eLife.87736.3.sa4

## Additional files

### Supplementary files
- MDAR checklist

### Data availability

Analysis code and associated (processed) data are available at GitHub (copy archived at *Yates, 2024*). That public repository contains links to the individual data files (https://doi.org/10.6084/m9.figshare. 26508136.v1). Raw data files are very large and are available upon request to either of the co-senior authors (JLY, ACH). Finally, we made use of Allen Institute data that was publicly available (https:// portal.brain-map.org/circuits-behavior/visual-coding-neuropixels).

The following dataset was generated:

| Author(s) | Year | Dataset title | Dataset URL | Database and Identifier |
|---|---|---|---|---|
| Rowley DP, Yates JL | 2024 | Marmoset Visual Cortex Preprocessed Treadmill Data Combined Supersessions | https://doi.org/10. 6084/m9.figshare. 26508136.v1 | figshare, 10.6084/ m9.figshare.26508136.v1 |

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
