## [Editor Report · eLife assessment]

This **important** work advances our understanding of the differences in locomotion-induced modulation in primate and rodent visual cortexes and underlines the significant contribution cross-species comparisons make to investigating brain function. The evidence in support of these differences across species is **convincing**. This work will be of broad interest to neuroscientists.

---

## [Referee Report · Reviewer #1 (Public review)]

More than ten years ago, it was shown that activity in the primary visual cortex of mice substantially increases when mice are running compared to when they are sitting still. This finding 'revolutionised' our thinking about visual cortex, turning away from it being a passive image processor and highlighting the influence of non-visual factors. The current study now for the first time repeats this experiment in marmosets. The authors find that in contrast to mice, marmoset V1 activity is slightly suppressed during running, and they relate this to differences in gain modulations of V1 activity between the two species.

Strengths

- Replication in primates of the original finding in mice partly took so long, because of the inherent difficulties with recording from the brain of a running primate. In fact one recent, highly related study on macaques looked at spontaneous limb movements as the macaque was sitting. The treadmill for the marmosets in the current study is a very elegant solution to the problem of running in primates. It allows for true replication of the 'running vs stationary' experiment and undoubtedly opens up many possibilities for other experiments recording from a head-fixed but active marmoset.

- In addition to their own data in marmoset, the authors run their analyses on a publicly available data set in mouse. This allows them to directly compare mouse and marmoset findings, which significantly strengthens their conclusions.

- Marmoset vision is fundamentally different from mouse vision as they have a fovea and make goal-directed eye movements. In this revised version of their paper, the authors acknowledge this and investigate the possible effect of eye movements and pupil size on the differences they find between running and stationary. They conclude that eye input does not explain all these differences.

Significance

The paper provides interesting new evidence to the ongoing discussion about the influence of non-visual factors in general, and running in particular, on visual cortex activity. As such, it helps to pull this discussion out of the rodent field mainly and into the field of primate research. The bigger question of *why* there are differences between rodents and primates remains still unanswered, but the authors do their best to provide possible explanations. The elegant experimental set-up of the marmoset on a treadmill will certainly add new findings to this issue also in the years to come.

---

## [Referee Report · Reviewer #2 (Public review)]

This work aims at answering whether activity in primate visual cortex is modulated by locomotion, as was reported for mouse visual cortex. The finding that the activity in mouse visual cortex is modulated by running has changed the concept of primary sensory cortical areas. However, it was an open question whether this modulation generalizes to primates.

To answer this fundamental question the authors established a novel paradigm in which a head-fixed marmoset was able to run on a treadmill while watching a visual stimulus on a display. In addition, eye movements and running speed were monitored continuously and extracellular neuronal activity in primary visual cortex recorded using high-channel-count electrode arrays. This paradigm uniquely permitted to investigate whether locomotion modulates sensory evoked activity in visual cortex of marmoset. Moreover, to directly compare the responses in marmoset visual cortex to responses in mouse visual cortex the authors made use of a publicly-available mouse dataset from the Allen Institute. In this dataset the mouse was also running on a treadmill and observing a set of visual stimuli on a display. The authors took extra care to have the marmoset and mouse paradigms as comparable as possible.

To characterize the visually driven activity the authors present a series of moving gratings and estimate receptive fields with sparse noise. To estimate the gain modulation by running the authors split the dataset into epochs of running and non-running which allowed them to estimate the visually evoked firing rates in both behavioral states.

Strengths:

The novel paradigm of head-fixed marmosets running on a treadmill while being presented with a visual stimulus is unique and ideally tailored to answering the question that the authors aimed to answer. Moreover, the authors took extra care to ensure that the paradigm in marmoset matched as closely as possible to the conditions in the mouse experiments such that the results can be directly compared. To directly compare their data the authors re-analyzed publicly available data from visual cortex of mice recorded at the Allen Institute. Such a direct comparison, and reuse of existing datasets, is another strong aspect of the work. Finally, the presented new marmoset dataset appears to be of high quality, the comparison between mouse and marmoset visual cortex is well done and the results and interpretation straightforward.

Weaknesses:

It is known that the locomotion gain modulation varies with layer in mouse visual cortex, with neurons in the infragranular layers expressing a diversity of modulations (Erisken et al. 2014 Current Biology). However, for the marmoset dataset the layer information was unfortunately not recorded, leaving this point open for future studies.

Nonetheless, the aim of comparing the locomotion induced modulation of activity in primate and mouse primary visual cortex was convincingly achieved by the authors. The results shown in the figures support the conclusion that locomotion modulates the activity in primate and mouse visual cortex differently. While mice show a profound gain increase, neurons in primate visual cortex show little modulation or even a reduction in response strength.

This work will have a strong impact on the field of visual neuroscience but also on neuroscience in general. It revives the debate of whether results obtained in the mouse model system can be simply generalized to other mammalian model systems, such as non-human primates. Based on the presented results, the comparison between the mouse and primate visual cortex is not as straightforward as previously assumed. This will likely trigger more comparative studies between mice and primates in the future, which is important and absolutely needed to advance our understanding of the mammalian brain.

Moreover, the reported finding that neurons in primary visual cortex of marmosets do not increase their activity during running is intriguing, as it makes you wonder why neurons in the mouse visual cortex do so. The authors discuss a few ideas in the paper which can be addressed in future experiments. In this regard it is worth noting that the authors report an interesting difference between the foveal and peripheral part of the visual cortex in marmoset. It will be interesting to investigate these differences in more detail in future studies. Likewise, while running might be an important behavioral state for mice, other behavioral states might be more relevant for marmosets and do modulate the activity of primate visual cortex more profoundly. Future work could leverage the opportunities that the marmoset model system offers to reveal new insights about behavioral related modulation in the primate brain.

---

## [Referee Report · Reviewer #3 (Public review)]

Prior studies have shown that locomotion (e.g., running) modulates mouse V1 activity to a similar extent as visual stimuli. However, it's unclear if these findings hold in species with more specialized and advanced visual systems such as nonhuman primates. In this work, Liska et al. leverage population and single neuron analyses to investigate potential differences and similarities in how running modulates V1 activity in marmosets and mice. Specifically, they discovered that although a shared gain model could describe well the trial-to-trial variations of population-level neural activity for both species, locomotion more strongly modulated V1 population activity in mice. Furthermore, they found that at the level of individual units, marmoset V1 neurons, unlike mice V1 neurons, experience suppression of their activity during running.

A major strength of this work is the introduction and completion of primate electrophysiology recordings during locomotion. Data of this kind were previously limited, and this work moves the field forward in terms of data collection in a domain previously inaccessible in primates. Another core strength of this work is that it adds to a limited collection of cross-species data collection and analysis of neural activity at the single-unit and population level, attempting to standardize analysis and data collection to be able to make inferences across species. In particular, the findings on how the primate peripheral and foveal V1 representations functionally relate to and differ from the mice V1 representations speak to the power of these cross-species comparisons.

However, there are still some lingering potential extensions to this work, largely acknowledged by the authors. One of these extensions involves more detailed eye movement analysis within species, such as microsaccades in marmosets and the potential impact on marmoset V1 activity. In the mouse data, similar eye-related analyses were not possible, in part due to instability in the eye recordings of many mouse sessions that made it challenging to replicate partnered analyses for the marmosets. We agree with the authors' assessment that these analyses can be targeted in future work and still believe that the marmoset eye-movement findings provide novel insights that will inform future cross-species comparisons of the visual system. Furthermore, another important issue not fully explored is the possible effects of the reward scheme during marmoset locomotion on V1 activity. The authors note that, unlike their mice counterparts, the marmosets were encouraged to run via liquid rewards, given after subjects traversed a specific distance. While the authors discuss the changes in arousal present when marmosets were running, there are still some unanswered questions on how their reward scheme may affect biomarkers (e.g., pupil sizes) and marmoset V1 activity.

Overall, the methods and data support the work's main claims. Single neuron and population level approaches demonstrate that the activity of V1 in mice and marmoset are categorically different. Since primate V1 is so diverse and differs from mouse V1, this presents important limitations on direct inferences from mouse V1 to primate V1. This work is a great step forward in the field, especially with the novel methodology of collecting neural activity from running primates.

---

## [Author Response]

The following is the authors’ response to the original reviews.

**Reviewer 1(Public Review):**
“but an obvious influencing factor that the authors could investigate in their own data set is the retinal input. In Fig1b, the authors even show these data in the form of gaze and pupil size. In these example data, by eye, it looks like the pupil size is positively correlated with the run speed. This would of course have large consequences on the activity in V1, but the authors do not do anything with these data. The study would improve substantially if the authors would correlate their run speed traces with other factors that they have recorded too, such as pupil size and gaze.”

Absolutely. We have added a first level of eye movement (and pupil size) analyses to the revised manuscript, resulting in an additional figure. In short, we found that eye movements are unlikely to play a significant role in our primary results, as the patterns of eye movements differed only slightly between running and stationary periods, and the measured impacts of such eye movements were also quantitatively much smaller than the primary effect sizes.

We also note that in analyzing the eye movements, we also found that pupil size was larger during running than stationary. This is suggestive evidence that running is correlated with increases in arousal. Although more work will be needed to calibrate and quantify how much this factor affects neural responses (and perhaps to dissociate it from running per se), the simple analysis we present suggest that the large differences we observe could be explained by a difference between how arousal and running are correlated in the monkey versus the mouse. Instead, it appears that both species have at least qualitatively similar relations between pupil size (a standard proxy for arousal) and running.

On this issue, we have added extensive discussion of the relevant recent work by Talluri et al. (2023) who attempted a similar cross-species analysis that considered spontaneous body movements and their effect on cortical activity (as well as the possibility that eye movements are a critical mediator in these modulations). Due to delays in revising our manuscript, we regret that our earlier submission had not cited this work, but we now do our best to highlight its importance and the synergy between these two papers. The full citation is listed below:

Talluri BC, Kang I, Lazere A, Quinn KR, Kaliss N, Yates JL, Butts DA, Nienborg H. Activity in primate visual cortex is minimally driven by spontaneous movements. Nat Neurosci. 2023 Nov;26(11):1953-1959. doi: 10.1038/s41593-023-01459-5.

There is a finer level of analysis that we hope to do in the future along these lines. It would rely on detailed characterization of each receptive field, building an image-computable model linking those receptive fields to the neural activity, and doing so at a finer time grain that links individual eye movements and changes in the spike train within a stimulus presentation (as opposed to working at the level of spike counts per stimulus presentation). Because these steps need to be accomplished together— and each requires substantial additional work and would go beyond the first-order findings we report in this work— we hope to report on such finer analyses in a standalone paper later. We are working on being able to do this in both marmoset and mouse.

More generally, we want to emphatically agree that what is missing from this paper is the “why?”! We have done our best to show that a fair comparison reveals quantitatively different phenomena in marmoset and mouse. In the revised discussion, we lay out many lines of work that we hope will gain traction on this deeper mechanistic point. There’s a lot to do, and several of the possibilities are already current topics of exploration in our ongoing work.

“Looking at the raster plot, however, shows that this strong positive correlation must be due entirely to the lower half of the neurons significantly increasing their firing rate as the mouse starts to run; in fact, the upper 25% or so of the neurons show exactly the opposite (strong suppression of the neurons as the mouse starts running). It would be more balanced if this heterogeneity in the response is at least mentioned somewhere in the text.”

We are also intrigued by the heterogeneity of effects at the single neuron level. That is why the next section of the paper is dedicated to analyzing effects on a cell-by-cell basis. The fractions of neurons showing either increases or decreases are described separately, to get at this very issue.

**Reviewer 2 (Public Review)::**
“For example, it is known that the locomotion gain modulation varies with layer in the mouse visual cortex, with neurons in the infragranular layers expressing a diversity of modulations (Erisken et al. 2014 Current Biology). However, for the marmoset dataset, it was not reported from which cortical layer the neurons are from, leaving this point unanswered.”Reviewer 2 called for more consideration of details that have been addressed in the mouse literature, such as the cortical layer of the cells, and related aspects of circuitry. We have greatly re-worked the Discussion to address several of these issues. In short, the manuscript’s set of data were collected without strong traction on layers or cell types, and it will be quite interesting to get a better handle on this using both refinements to our recording procedures as well as new techniques that are now possible in the marmoset for future studies.“In this regard, it is worth noting that the authors report an interesting difference between the foveal and peripheral parts of the visual cortex in marmoset. It will be interesting to investigate these differences in more detail in future studies. Likewise, while running might be an important behavioral state for mice, other behavioral states might be more relevant for marmosets and do modulate the activity of the primate visual cortex more profoundly. Future work could leverage the opportunities that the marmoset model system offers to reveal new insights about behavioral-related modulation in the primate brain.”

Same page! We have expanded the discussion to better emphasize these points and are already deep in follow up experiments to explore the foveal and peripheral representations.

**Reviewer 3(Public Review):****:**
“However, the authors did not take full advantage of the quantity and diversity of the marmoset visual cortex recordings in their analyses. They mention recording and analyzing the activity of peripheral V1 neurons but mainly present results involving foveal V1 neurons. Foveal neurons, with their small receptive fields strongly affected by precise eye position, would seem to be less likely to be comparable to rodent data. If the authors have a reason for not doing so, they should provide an explanation.”

We agree, and hope the reviewer finds our overall reply, detailed response to Reviewer 1 (who raised a similar issue), and corresponding updates to the manuscript appropriate for this stage of understanding.

“Given that the marmosets are motivated to run with liquid rewards, the authors should provide more context as to how this may or may not affect marmoset V1 activity. Additionally, the lack of consideration of eye movements or position presents a major absence for the marmoset results, and fails to take advantage of one of the key differences between primate and rodent visual systems - the marmosets have a fovea, and make eye movements that fixate in various locations on the screen during the task.”

In addition to the response above, we have made edits to the manuscript to speak to issues of arousal and eye movements (also detailed in previous responses). Given the modest decrease in activity we see, the usual concerns about potential increases in neural activity related to eye movements (which we quantify in the revision) and other issues related to motivation are hard to specifically relate to existing literature. But in the revised Discussion we talk more about how future work can/should dissociate these factors, as has been done in the mouse literature.

“Finally, the model provides a strong basis for comparison at the level of neuronal populations, but some methodological choices are insufficiently described and may have an impact on interpreting the claims.”

We have also clarified the shared-gain model’s description, which we agree needed additional detail and clarification.